# LEAVES: LEARNING VIEWS FOR TIME-SERIES DATA IN CONTRASTIVE LEARNING

## ABSTRACT

Contrastive learning, a self-supervised learning method that can learn representations from unlabeled data, has been developed promisingly. Many methods of contrastive learning depend on data augmentation techniques, which generate different views from the original signal. However, tuning policies and hyper-parameters for more effective data augmentation methods in contrastive learning is often time and resource-consuming. Researchers have designed approaches to automatically generate new views for some input signals, especially on the image data. But the view-learning method is not well developed for time-series data. In this work, we propose a simple but effective module for automating view generation for time-series data in contrastive learning, named **lea**rning **vie**ws for time-**s**eries data (LEAVES). The proposed module learns the hyper-parameters for augmentations using adversarial training in contrastive learning. We validate the effectiveness of the proposed method using multiple time-series datasets. The experiments demonstrate that the proposed method is more effective in finding reasonable views and performs downstream tasks better than the baselines, including manually tuned augmentation-based contrastive learning methods and SOTA methods.

## 1 INTRODUCTION

Contrastive learning has been widely applied to improve the robustness of the model for various downstream tasks such as images (Chen et al., 2020; Grill et al., 2020; Wang & Qi, 2022) and time-series data (Mohsenvand et al., 2020; Mehari & Strodthoff, 2022). Among the developed contrastive learning methods, data augmentation plays an essential role in generating different corrupted transformations, as views of the original input for the pretext task. For example, Chen et al. (2020) proposed a SimCLR method to maximize the agreements of augmented views from the same sample to pre-train the model, which significantly outperformed the previous state-of-the-art method in image classification with far less labeled data. However, the data augmentation methods selection is usually empirical, and tuning a set of optimized data augmentation methods can cost thousands of GPU hours even with the automating searching algorithms (Cubuk et al., 2019). Therefore, it remains an open question how to effectively generate views for a new dataset.

Instead of using artificially generated views, researchers have been putting efforts into training deep learning methods to generate optimized views for the input samples (Tamkin et al., 2020; Rusak et al., 2020). These methods generate reasonably-corrupted views for the image datasets and result in satisfactory results. For example, Tamkin et al. (2020) proposed the ViewMaker, an adversarially trained convolutional module in contrastive learning, to generate augmentation for images. Nevertheless, the aforementioned method, such as the ViewMaker, might not be acclimatized when forthrightly utilized on the time-series data. The main challenge is that, for the time-series signal, we need to not only disturb the magnitudes (spatial) but also distort the temporal dimension Um et al. (2017); Mehari & Strodthoff (2022). While the image-based methods can only disturb the spatial domain by adding reasonable noise to the input data.

In this work, we propose LEAVES, which is a lightweight module for learning views on time-series data in contrastive learning. The LEAVES is optimized adversarially against the contrastive loss to generate challenging views for the encoder in learning representations. In addition, to introduce smooth temporal perturbations to the generated views, we propose a differentiable data augmentation technique for time-series data, named *TimeDistort*. Figure 1 shows the examples of the gener-

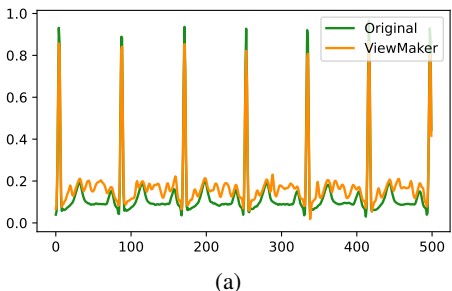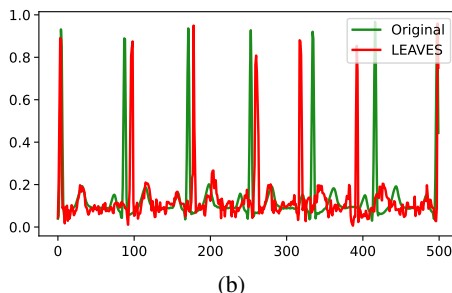

(a)                                    (b)

Figure 1: Visualization of the views learned using (a) the ViewMaker and (b) the proposed LEAVES. Compared to the ViewMaker, LEAVES introduces temporal distortion and the augmented view is more faithful.

ated views of electrocardiogram (ECG) from the ViewMaker (Tamkin et al., 2020) and our method. We can find that no temporal location is perturbed in Fig.1 (a), and the flat region (T-P interval as the ECG fiducial) of the original ECG signal was completely distorted. Compared to the View-Maker, the proposed LEAVES can distort the original signal in both spatial and temporal domains, more importantly, reducing the risk of losing intact information due to excessive perturbation in time-series data. Our experiments and analysis showed that the proposed LEAVES (1) outperforms the baselines, including SimCLR and the SOTA methods, and (2) generates more reasonable views compared to the SOTA methods in time-series data.

## 2   RELATED WORKS

### 2.1   AUGMENTATION-BASED CONTRASTIVE LEARNING

Among the contrastive learning algorithms proposed in various areas, the data augmentation methods usually play essential roles in generating views from the original input to form contrastive pairs. Many contrastive learning frameworks have been recently developed based on the image transformation in computer vision (He et al., 2020; Chen et al., 2020; Grill et al., 2020; Chen & He, 2021; Tamkin et al., 2020; Zbontar et al., 2021; Wang & Qi, 2022; Zhang & Ma, 2022). For example, Chen et al. (2020) proposed a SimCLR framework that maximizes the agreement between two views transformed from the same image. BYOL (Grill et al., 2020) encourages two networks, including a target network and an online network, to interact and learn from each other based on two augmented views of an image. Zbontar et al. (2021) proposed a Barlow Twin framework that applied two corrupted views from an image with a redundancy-reduction objective function to avoid trivial constant solutions in contrastive learning.

Other than the applications in the computer vision area, contrastive learning algorithms have also been applied to time-series data (Gopal et al., 2021; Mehari & Strodthoff, 2022; Wickstrøm et al., 2022). For example, Gopal et al. (2021) proposed a clinical domain-knowledge-based augmentation on ECG data and generated views from ECG from contrastive learning. Also, Mehari & Strodthoff (2022) applied well-evaluated methods such as SimCLR, BYOL, and CPC Oord et al. (2018) on time-series ECG data for clinical downstream tasks. Wickstrøm et al. (2022) generated contrastive views by applying the MixUp augmentation (Zhang et al., 2017) in the time-series data. The aforementioned research has achieved promising results by leveraging unlabeled data, however, the empirically augmented views might not be optimal, especially for the datasets that are relatively new or less popular, as exploring appropriate sets of augmentations is expensive.

### 2.2   AUTOMATIC AUGMENTATION

Rather than setting augmentation methods with empirical settings, researchers proposed multiple methods of optimizing the appropriate augmentation strategies (Cubuk et al., 2019; Ho et al., 2019; Lim et al., 2019; Li et al., 2020; Cubuk et al., 2020; Liu et al., 2021). For example, AutoAugment (Cubuk et al., 2019) was designed as a reinforcement learning-based algorithm to search for augmentation policies, including the possibility and order of using different augmentation methods. DADA (Li et al., 2020) is a gradient-based optimization strategy for finding the augmentation policy with the highest probabilities after training, which significantly reduces the search time compared to

algorithms such as AutoAugment. These searching methods have been proven to have high performance, however, they usually require heavy computational complexity in thoroughly searching the augmentation space and finding optimized policies.

Instead of searching for augmentations from the policy space, researchers also developed learning views, which can be understood as generating data transformation by neural network rather than using manually tuned augmentations (Tian et al., 2020; Rusak et al., 2020; Tamkin et al., 2020). For example, Rusak et al. (2020) applied a CNN structure to generate noise based on the input data and trained the perturbation generator in an adversarial manner against the supervised loss. Similarly, Tamkin et al. (2020) proposed a ResNet-based ViewMaker module to generate views for data for the contrastive learning framework. The training of ViewMaker was also adversarial by maximizing the contrastive loss against the objective of the representation encoder. Nevertheless, these methods lack consideration of temporal perturbation when used in time-series datasets. Therefore, in this work, we design our proposed LEAVES module to generate both the magnitude and temporal perturbations in sequences.

## 3 METHODS

Contrastive learning is a self-supervised learning approach, which encourages the representations of transformations of the same inputs to be similar and learns the differences from pairs of different samples. In this work, we adopt a simple and well-proven contrastive learning method, the SimCLR (Chen et al., 2020). Figure 2 shows the overview of the pre-training architecture. We first introduce a differentiable LEAVES module, which can generate more challenging but still faithful views of an input. The LEAVES module is then plugged into the SimCLR framework to generate different views for contrastive learning. The LEAVES is trained with the encoder in an adversarial manner.

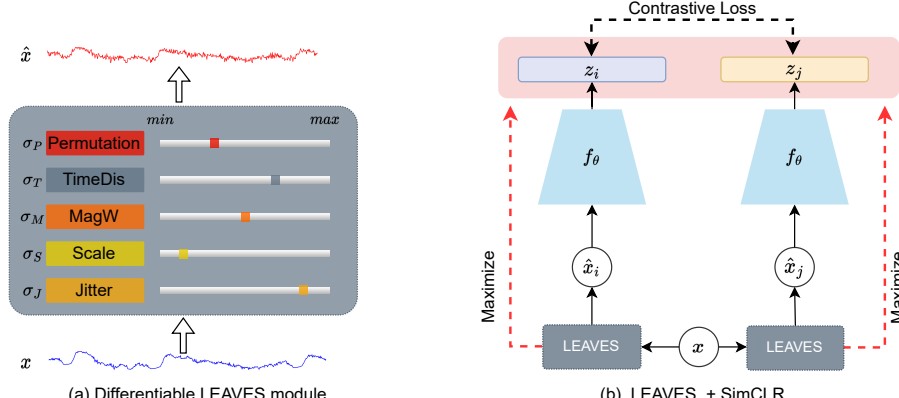

(a) Differentiable LEAVES module  (b) LEAVES + SimCLR

Figure 2: LEAVES is designed as a plug-in module to generate augmented views in time-series data for contrastive learning. The framework trains the LEAVES and the representation encoder in an adversarial manner: LEAVES minimizes the agreements between the pairs of representations, whereas the encoder is optimized to maximize the agreement.

### 3.1 LEAVES

We propose a LEAVES module, which is a lightweight component that can be easily plugged into the contrastive learning system. This module consists of a series of differentiable data augmentation methods, including *Jitter* $T_J$, *Scale* $T_S$, magnitude warping (*MagW*) $T_{MW}$, permutation (*Perm*) $T_P$, and a newly proposed method named time distortion (*TimeDis*) $T_{TD}$. To simplify the expression, we use $\odot$ to indicate the in-series operation. For example, $T_J \odot T_P$ represents that the input data has been first transformed by adding jittering noise, then been permuted. Thus, the proposed module generates view $\hat{X}$ by

$$\hat{\mathbf{X}} = \mathbf{X} \odot T_J(\sigma_J) \odot T_S(\sigma_S) \odot T_{MW}(\sigma_M) \odot T_{TD}(\sigma_T) \odot T_P(\sigma_P) \tag{1}$$

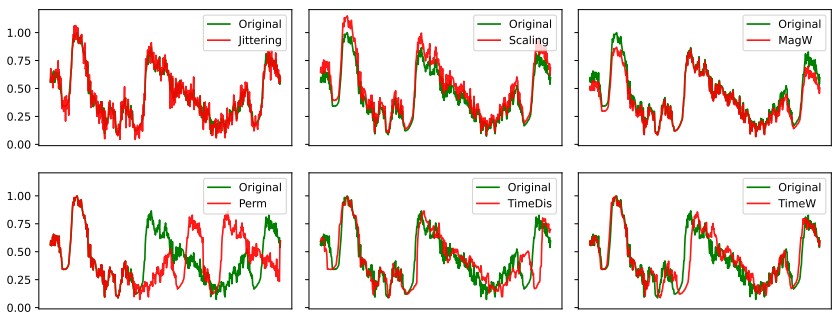

Figure 3: Examples of different data augmentation methods for time-series data

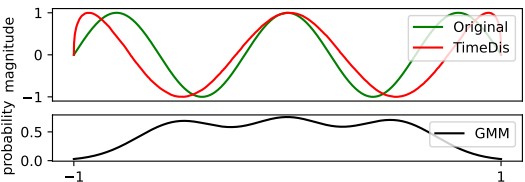

Figure 4: Example of the time distortion augmentation with 3 Gaussian components. Corresponding locations with higher probabilities in the generated GMM are up-sampled in the affine space. Conversely, locations with smaller probabilities are down-sampled in the transformation.

where $\sigma$ represents the hyper-parameters of the data augmentation method that control the intensity of corruption on the original sample. For instance, $\sigma_J$ represents the standard deviation value for generating the jittering noise. The target learning parameters of this module are the $\sigma$ of augmentation methods. By learning these parameters, the module learns the strategy of generating views by combining multiple augmentation methods. We do not deliberately tune the order of augmentations applied to $X$ in equation 1, as the hyper-parameters and views of augmentations are independent, e.g., the applied *Scale* operation would not rely on the view generated by *Jitter*.

### 3.1.1 DIFFERENTIABLE DATA AUGMENTATIONS FOR TIME-SERIES DATA

Several widely used data augmentation methods are selected for LEAVES. For example, *Jitter, Scale*, and *MagW* perturb the magnitude of the original signal; whereas time warping (*TimeW*) (Um et al., 2017) and *Perm* corrupt the temporal location. A detailed description of the augmentation methods can be found in Appendix A.1.

To optimize the hyper-parameters in these augmentation algorithms, the gradients are needed to be propagated back to these parameters during the training process. However, these augmentation methods are based on non-differentiable operations such as drawing random values, indexing, etc. Therefore, we applied reparameterization tricks (Jang et al., 2016; Maddison et al., 2016) to make those procedures differentiable, except the *TimeW* method, as the indexing operation makes it difficult to retrieve gradients. Thus, we propose the *TimeDis* augmentation method as an alternative to smoothly distort temporal information (see details in the next subsection). Figure 3 shows the examples of the six augmentation methods on a time-series sample. To ensure the generated corruptions are reasonable, we constrain the noise with an up-bound $\eta$ as the maximum values of $\sigma$ values in magnitude-based methods *Jitter, Scale, MagW*, and $K$ as the maximum segments in *Perm*.

**TimeDis**. This method relies on a smooth probability distribution to generate the probability of the location to be sampled in the original signal. We utilize a reparameterized Gaussian mixture model with $M$ Gaussian components as $\sum_i^M \phi_i \mathcal{N}(\mu_i, \sigma_i^2)$, to generate the location indexes $\lambda \in \mathbb{R}^{N \times C \times L}$ from -1 to 1. Figure 4 shows an example of *TimeDis*. Among the generated location indexes, -1 corresponds to the first time step (position 1) in the original signal, whereas 1 corresponds to the last time step (position $L$). Then, we use $\lambda$ to affine the original signal $X$ as the view $\hat{X}$, which results that for the locations with dense indexes in $\lambda$, the intervals among samples become looser; whereas for these locations with sparse indexes in $\lambda$, the corresponding intervals would be tighter.

## 3.2 Adversarial Training

In representation learning, we define the representation extracted by the encoder as $z$. Given $N$ pairs of representations extracted by the encoder in the SimClR framework as $(z_i, z_j)$, $\{i, j\} \in [1, N]$, the loss function of maximizing the agreements between pairs of views can be defined as:

$$\mathcal{L} = \frac{1}{2N} \sum_{k=1}^{N} [\ell(2k-1, 2k) + \ell(2k, 2k-1)], \tag{2}$$

$$\ell_{i,j} = -\log \frac{\exp(s(z_i, z_j)/\tau)}{\sum_{k=1}^{2N} \mathbb{1}_{k \neq i} \exp(s(z_i, z_k)/\tau)} \tag{3}$$

where $s(z_i, z_j)$ represents the cosine similarity between $z_i$ and $z_j$, $\mathbb{1}_{k \neq i}$ is an indicator function that equals to 1 *iff* $k \neq i$, and $\tau$ denotes a temperature parameter that is fixed as 0.05 in this study.

As shown in Figure 2, the LEAVES and the encoder are optimized in opposite directions: the target objective of the encoder is to minimize $\mathcal{L}$, whereas the LEAVES module desires to maximize $\mathcal{L}$. By leveraging the adversarial training manner, we design the LEAVES module to distort the original signal as challenging as possible while the encoder can still extract intact information from the view pairs. Under this scenario, the encoder would be robust by training against the most corrupted views. After training the SimCLR framework, the learned model weights in the encoder structure are used to initialize the model weights for supervised learning in downstream tasks.

## 4 Evaluation

To evaluate the proposed method, we conduct experiments on 4 public times-series datasets, including Apnea-ECG (Penzel et al., 2000), Sleep-EDFE (Kemp et al., 2018), PTB-XL (Wagner et al., 2020), and PAMAP2 (Reiss & Stricker, 2012) for applications of detecting apnea, sleep stage, arrhythmia, and human activities, respectively. The detailed information of the used datasets can be found in Appendix B. For each of the datasets, we pre-train the proposed module and the encoder and fine-tune the encoder for the downstream task. Three baselines are implemented for comparison, including (1) supervised ResNet-18, (2) SimCLR with random augmentations (see fine-tuning details in 5.2), and (3) reproduced ViewMaker network that intakes one-dimension time-series inputs.

### 4.1 Sleep Apnea Detection with Single-Lead ECG

The Apnea-ECG dataset (Penzel et al., 2000) studies the relationship between human sleep apnea symptoms and heart activities (monitored by ECG), which can be accessed through Physionet(Goldberger et al., 2000). Following the settings in the original release Penzel et al. (2000), we used a one-minute scale of 100Hz ECGs to detect the binary labels of whether apean occurs, with 17233 and 17010 samples in the training and testing sets, respectively. In the contrastive pre-training stage, we set the noise threshold $\delta$ to be 0.05 for *Jitter, Scale, MagW*, $M$ as 12 for *TimeDis*, and $K$ as 5 for *Perm*. We used 100 epochs with a learning rate of 1e-3 to pre-train the encoder and 30 epochs with a learning rate of 1e-3 to fine-tune for the downstream task.

Table 1 shows the evaluation results for detection sleep apnea. Following the SOTAs on the same dataset, we used sensitivity (Sen.) and specificity (Spec.), which assess the ability of the model to diagnose apnea for patients, as metrics to measure the model performances. When comparing the proposed model with the baseline models, we observed that LEAVES performed better than the baselines in both metrics of Sen. and Spec. And both SimCLR and ViewMaker outperformed the supervised baseline, with SimCLR showing slightly higher performances than the ViewMaker. Compared to the SOTAs, we found the proposed method showed a competitive Sen. score but a relatively low Spec. score. This can be caused by different settings, such as filtering out noisy samples and data pre-processing procedures. The baseline supervised structure is similar to in (Chang et al., 2020); however, our resulted Spec. in supervised baseline was lower than SOTAs.

Table 1: Evaluation results in detecting sleep apnea on the Apnea-ECG dataset. The bold values represent the best performance among the baseline models and the proposed methods.

| | Methods | Sen. | Spec. |
|---|---|---|---|
| SOTAs | Multi-Scale ResNet (Fang et al., 2022) | 0.841 | 0.871 |
| | 1D CNN (Chang et al., 2020) | 0.811 | 0.920 |
| | CNN + Decision Fusion (Singh & Majumder, 2019) | 0.900 | 0.838 |
| | Auto-Encoder + HMM (Li et al., 2018) | 0.889 | 0.821 |
| Baselines | Supervised | 0.843 | 0.788 |
| | SimCLR | 0.856 | 0.803 |
| | ViewMaker | 0.851 | 0.795 |
| Ours | LEAVES | **0.865** | **0.821** |

Table 2: Evaluation results in classifying sleep stages using EEG on the Sleep-EDF dataset. The bold values represent the best performance among the baseline models and the proposed methods.

| | Methods | Acc. | F1 |
|---|---|---|---|
| SOTAs | U-Time (1D CNN-based) (Perslev et al., 2019) | - | 0.764 |
| | CNN + RNN (Mousavi et al., 2019) | 84.3 | 0.796 |
| | U-Sleep (1D CNN-based) (Perslev et al., 2021) | - | 0.790 |
| | 1D CNN + HMM (Li et al., 2018) | 84.0 | 0.78 |
| | Spectrogram + Transformer Phan et al. (2022) | 84.9 | 0.789 |
| Baselines | Supervised | 83.6 | 0.769 |
| | SimCLR | 84.8 | 0.783 |
| | ViewMaker | 84.4 | 0.777 |
| Ours | LEAVES | **85.3** | **0.790** |

## 4.2 SLEEP STAGE CLASSIFICATION WITH EEG

Electroencephalography (EEG) is an essential signal to monitor human brain activities. We tested our methods on the Sleep-EDF (expanded) (Kemp et al., 2018) dataset, which contains whole-night sleep recordings of 100Hz Fpz-Cz EEG signal. Following Supratak & Guo (2020), we extracted 42308 30-second samples that were annotated in 5 sleep stages. In the contrastive pre-training stage, we set the noise threshold $\delta$ to be 0.05 for *Jitter, Scale, MagW*, $M$ as 10 for *TimeDis*, and $K$ as 5 for *Perm*. We used 100 epochs with a learning rate of 1e-3 to pre-train the encoder and 30 epochs with a learning rate of 1e-3 to fine-tune for the downstream task. To compare the model performances with SOTAs, we use accuracy and macro f1-score as the metrics for evaluation.

Table 2 shows the performances of classifying sleep stages using EEG signal. From the table, we can learn that the proposed methods outperformed the baselines. Also, when comparing our results with SOTAs, we found the achieved performances in both accuracy and macro f1-score were competitive. However, we need to admit that the different experimental settings were used in the SOTAs and our work. There were several factors existed that might trigger discrepancies in performances. For example, the pre-processing procedures were not uniform; also, the split train/test datasets were not universal, i.e., 10 or 20-fold cross-validations with different settings were widely applied in SOTAs, whereas we split validation sets according to the subject IDs.

## 4.3 HUMAN ACTIVITIES DETECTION WITH IMU AND HEART RATE

Human activities can be detected using wearable device data. The PAMAP2 (Reiss & Stricker, 2012) studies the relationship between human activities and data collected from wearable sensors of 3 inertial measurement units (IMU) and a heart rate monitor. We used 100-Hz IMU data and the upsampled heart rate data in the experiments. Following Moya Rueda et al. (2018); Tamkin et al. (2020), 12 of the total 18 different physical activities are used in the experiments. In the contrastive pre-training stage, we set the noise threshold $\delta$ to be 0.05 for *Jitter, Scale, MagW*, $M$ as 7 for *TimeDis*, and $K$ as 5 for *Perm*. We used 100 epochs with a learning rate of 1e-3 to pre-train the encoder and 20 epochs with a learning rate of 1e-3 to fine-tune for the downstream task. To compare the model performances with SOTAs, we used accuracy and macro f1-score as the metrics for evaluation.

Table 3: Evaluation results in detecting human activities on PAMAP2 dataset. The bold values represent the best performance among the baseline models and the proposed methods.

| | Methods | Acc. | F1 |
|---|---|---|---|
| SOTAs | 1D CNN (Moya Rueda et al., 2018) | 93.7 | 0.937 |
| | ViewMaker (Spectrogram + CNN) (Tamkin et al., 2020) | 91.3 | - |
| | Late-Fusion 1D CNN (Mekruksavanich & Jitpattanakul, 2021) | 85.5 | - |
| | Multi-Input CNN-GRU Dua et al. (2021) | 95.3 | 0.952 |
| | (subject-dependent) ResNet + BiLSTM (Li & Wang, 2022) | 97.2 | 0.974 |
| Baselines | Supervised | 89.9 | 0.896 |
| | SimCLR | 92.2 | 0.925 |
| | ViewMaker | 91.5 | 0.913 |
| Ours | LEAVES | **93.3** | **0.934** |

Table 4: Evaluation results in detecting cardiac arrhythmia on PTB-XL dataset. The bold values represent the best performance among the baseline models and the proposed methods.

| | Methods | AUC | Acc. |
|---|---|---|---|
| SOTAs | 1D CNN (Śmigiel et al., 2021) | 0.910 | 76.5 |
| | Siwn Transformer (Li et al., 2021) | 0.905 | - |
| | Multi-Lead-Branch Fusion CNN (Zhang et al., 2021) | 0.943 | - |
| | CPC (Mehari & Strodthoff, 2022) | 0.941 | - |
| | SimCLR (Mehari & Strodthoff, 2022) | 0.926 | - |
| | BYOL (Mehari & Strodthoff, 2022) | 0.926 | - |
| Baselines | Supervised | 0.924 | 77.3 |
| | SimCLR | **0.932** | **79.5** |
| | ViewMaker | 0.926 | 78.0 |
| Ours | LEAVES | 0.931 | 79.3 |

Table 3 shows the performances in classifying human activities with PAMAP2 dataset. Our proposed method outperformed all the baselines and showed competitive results as SOTAs that shared the same train/test settings (Moya Rueda et al., 2018; Tamkin et al., 2020). The study conducted by Li & Wang (2022) used a 70/30% train/test split strategy in a subject-dependent setting and achieved the highest performance among all studies. In addition, Table 3 shows the comparison of the results reported in the original work (Tamkin et al., 2020) and the 1D version we reproduced. The model accuracy from the original work, which converted time-series data into spectrograms and leveraged 2D ResNet, is very similar to the accuracy from our implemented 1D ResNet version.

## 4.4 ARRHYTHMIA CLASSIFICATION WITH 12-LEAD ECG

Cardiac arrhythmia is one of the main causes of cardiovascular diseases, thus detecting arrhythmia has important clinical prospects. The PTB-XL(Wagner et al., 2020) dataset is a large dataset with contains 21,837 12-lead and 10-second ECG in 100 Hz with arrhythmia labels in 5 classes. We follow the recommended splits of training and testing sets in the original work (Wagner et al., 2020). In the contrastive pre-training stage, we set the noise threshold $\delta$ to be 0.05 for *Jitter, Scale, MagW*, $M$ as 6 for *TimeDis*, and $K$ as 5 for *Perm*. We used 100 epochs with a learning rate of 1e-3 to pre-train the encoder and 30 epochs with learning late of 1e-3 to fine-tune for the downstream task. To compare the model performances with SOTAs, we use AUC and accuracy as the metrics for evaluation.

Table 4 shows the results of classifying arrhythmia with ECG sequences. We found the proposed method outperformed the supervised and ViewMaker baselines; whereas the SimCLR baseline with random augmentations performed slightly higher than the proposed method. When comparing our method to the SOTAs, we found that our method showed higher AUC than some supervised methods such as in (Śmigiel et al., 2021; Li et al., 2021). Interestingly, when comparing our results in an ECG-centered benchmark self-supervised learning work, it shows that our results were slightly higher than their implemented SimCLR, which might also indicate the effectiveness of the proposed LEAVES compared to manually tuned augmentations (Mehari & Strodthoff, 2022).

Table 5: The quality check results on the learned views in the Apnea-ECG dataset

|  | Excellent | Barely Acceptable | Unacceptable |
|---|---|---|---|
| Original Data | 4.4% | 92.4% | 3.2% |
| ViewMaker | 0.1% | 50.4% | 49.5% |
| (Ours) LEAVES | 0.4% | 96.0% | 3.6% |

Table 6: The results table of tuning augmentation methods in the SimCLR framework. The performances are listed in the metrics of accuracy/macro f1-score

|  | $T(0.01)$ | $T(0.02)$ | $T(0.03)$ | $T(0.04)$ | $T(0.05)$ |
|---|---|---|---|---|---|
| Apnea-ECG | 76.1/0.752 | 76.0/0.746 | **78.8/0.775** | 76.5/0.759 | 78.5/0.771 |
| Sleep-EDFE | 82.5/0.771 | 83.2/0.775 | 84.0/0.778 | **84.8/0.783** | 84.5/0.780 |
| PTB-XL | 76.7/0.663 | 78.7/0.670 | **79.5/0.676** | 77.6/0.671 | 75.1/0.649 |
| PAMAP2 | 89.0/0.897 | 90.5/0.901 | 91.1/0.912 | **92.2/0.925** | 91.3/0.918 |

## 5 DISCUSSION

This section introduces the ablation studies we conducted, including applying the ViewMaker framework in time-series data and fine-tuning the baselines SimCLR algorithm with different augmentation hyper-parameters. Also, we further introduce the proposed method's learning views and the complexity of the LEAVES module.

### 5.1 ABLATION STUDY: VIEWMAKERS IN TIME-SERIES DATA

As stated in the previous section, our work is inspired by the ViewMaker (Tamkin et al., 2020). We tested the ViewMaker framework, and improvements were observed as shown in the evaluation section. However, we also observed limitations of the ViewMaker while applying it to the time-series data. Figure 1 shows an example of its limitation in temporal distortion and preserving intact information as mentioned in section 1. To further examine the faithfulness of the generated views, we utilized an ECG quality check method (Zhao & Zhang, 2018) using the NeuroKit package (Makowski et al., 2021). Table 5 shows the quality check results on Apnea-ECG dataset. We can observe that the ViewMakers method perturb almost half of the ECGs into "Unacceptable", representing the signals that are hardly recognized as ECG signals. These limitations of ViewMaker when applied to the time-series dataset motivated us to develop the proposed method in this work.

### 5.2 ABLATION STUDY: FINE-TUNE THE SIMCLR BASELINE

As mentioned in the previous sections, finding optimal data augmentation methods in contrastive learning is challenging as the search space of the augmentation methods is usually massive. In this work, to train a strong SimCLR baseline, we tuned the intensity of the augmentation methods for the baseline SimCLR mentioned in section 4. To simplify the experiments, we controlled the standard deviation $\sigma$ of *Jitter, Scale, MagW, TimeW* to be the same value, increasing from 0.01 to 0.05 by 0.01, and fixed the maximum permutation segment as 5. For example, $T(0.01)$ represents that the $\sigma$ values of *Jitter, Scale, MagW, TimeW* are 0.01 and $K = 5$ for permutation. We listed the tuning performances with metrics of accuracy and macro f1-score in Table 6. With different hyperparameters that affect the intensity of the augmentation methods, we observed different evaluation results. On some datasets, such as PAMAP2, we observed close performance while using different hyperprameters. However, the model performances seemed to be more hyper-parameter-sensitive on the PTB-XL dataset. For example, while using the $\sigma$ values of 0.05, we observed a performance drop by a big margin compared to the SimCLR with $\sigma = 0.03$. This illustrated that finding reasonable augmentations makes contributions to contrastive learning, while on the other hand, inappropriate transformations may hurt the model performances. And our proposed method can find the appropriate augmentation for time series data without spending time in augmentation searching, which could be meaningful to researchers using new or not commonly studied time-series datasets.

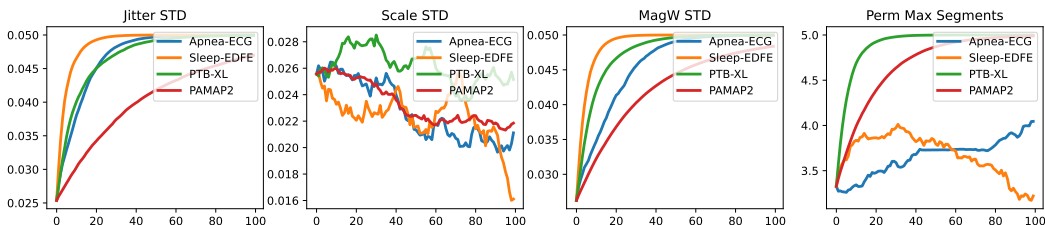

Figure 5: Visualization of the scalar hyper-parameters learned through LEAVES

### 5.3 LEARNING HYPER-PARAMETERS FOR AUGMENTATIONS

Given that we proposed a differentiable augmentation-based approach, we can speculate that the hyper-parameters controlling augmentation would change with the training process as the model learns. Figure 5 shows the scalar hyper-parameters change for augmentations of *Jitter, Scale, MagW*, and *Perm*. We can observe that the $\sigma$ values for *Jitter* and *MagW* were keeping increasing on all four dataset; whereas the $\sigma$ value for *Scale* showed a decreasing trend. We found the maximum segment $K$ in *Perm* showed an increasing trend on PTB-XL but decreased on PAMAP2 dataset. Although it was not conducted in this work, this phenomenon shows the possibility that our approach may help find appropriate views for supervised learning on the different datasets as well. A potential application is to combine our proposed module with a supervised learning framework by adversarial training, such as the framework by Rusak et al. (2020).

### 5.4 TIME & SPACE COMPLEXITIES

Since the targeted optimizing weights of LEAVES are the hyper-parameters of the augmentation methods, the proposed method shows advantages in terms of model space complexity compared to the previous SOTAs such as the ViewMaker. For example, there are 580K parameters to learn in our re-produced 1D ViewMaker structure; whereas LEAVES optimizes 20 parameters to generate views. On the other hand, the latency of introducing LEAVES in SimCLR can be negligible. Under the training environment of *AWS p3.2xlarge* (dual NVIDIA V100 GPUs) with the batch size $N$ of 128 for 100 training epochs on Sleep-EDFE dataset, the baseline SimCLR takes 578.0 seconds per epoch on average; whereas the SimCLR with LEAVES takes 390.8 seconds per epoch. Because augmentations are programmed as part of the model in LEAVES, which leverages GPU to accelerate computations, thus resulting in an even shorter training time than the baseline SimCLR.

## 6 CONCLUSION

In this work, we introduced a simple but effective LEAVES module to learn augmentations for time-series data in contrastive learning. With an adversarial training manner, the proposed method optimizes the hyper-parameters for data augmentation methods in contrastive learning. The proposed method was evaluated on 4 datasets, and the results showed performance improvements from the baselines. Particularly, without hyper-parameter tuning, the proposed LEAVES outperformed the SimCLR baseline on 3 out of 4 downstream applications. We also demonstrated the advantages of using our method in terms of preserving intact information in augmented views, especially on ECG time-series data, compared to the SOTA work. In the future, we will introduce more augmentation methods to LEAVES to improve the variability of the module and explore its potential in tuning augmentations in supervised learning. Also, we will apply our method to a broader range of time-series data. Further, investigating the interpretability of LEAVES is also an interesting direction for us to better understand the data augmentation policies in contrastive learning.

#### ACKNOWLEDGMENTS

Use unnumbered third-level headings for the acknowledgments. All acknowledgments, including those to funding agencies, go at the end of the paper.

REPRODUCIBILITY STATEMENT

To ensure the reproducibility of our work, we described the implementation details of the model and conducted evaluation experiments on publicly accessible datasets. We followed the procedures in the previous studies to process the data and stated the pre-processing details in Appendix. In addition, we uploaded our example codes as a part of the supplemental materials at this stage. Upon the acceptance of the manuscript, we will upload our work to a GitHub repository.

ETHICS STATEMENT

We conducted experiments and evaluated our proposed methods on public time-series datasets, including the data collected from human subject studies. The proposed LEAVES has shown its ability to help improve the contrastive learning model performance in clinical applications, such as detecting sleep apnea, arrhythmia, etc., by automatically tuning parameters for data augmentations without any requirements in domain knowledge. In addition, the proposed method can learn and generate different augmentations for samples, which can be considered an advantage in diversifying training sets in clinical applications and improving model generalizability, especially when researchers have limited but valuable data collected from study participants.

However, similarly to all other data-driven solutions in clinical studies, there exist risks in our method of making biased predictions on clinical applications. For example, the generated distortion on clinical time-series samples may bias the intact information for certain clinical usage. For example, the temporal location changes in ECG may distort information such as the heart rate of patients. Moreover, other factors, such as poor data quality, heterogeneity among different study participants, and annotation biases, may lead to inaccurate predictions. Thus, researchers should be aware of the limitations of our approach.

Moreover, all datasets used in this work are anonymized. The experiments were conducted using the datasets from PhysioNet, following the PhysioNet privacy policy [link] strictly. We ensure the datasets are not associated with any identifiable personal information from participants.

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

# A   EXTRA INFORMATION OF DATA AUGMENTATIONS

## A.1   DATA AUGMENTATION FOR TIME SERIES DATA

**Jittering** is a method of adding randomly generated noise to the original signal, and the generated noise obeys the Normal distribution:

$$\hat{\mathbf{X}} = \mathbf{X} + \epsilon_J, \quad \epsilon_J \sim \mathcal{N}(0, \sigma_J^2) \tag{4}$$

where $\epsilon_J \in \mathbb{R}^{N \times C \times L}$ represents the sampled noises. $\sigma_J$ is the hyper-parameter that needed to be determined.

**Scaling** is to change amplitude of the signal by multiply the signal with random factors by each signal channel of the data:

$$\hat{\mathbf{X}} = [\hat{x}_1, \hat{x}_2, ..., \hat{x}_C] = [\epsilon_S^1 x_1, \epsilon_S^2 x_2, ..., \epsilon_S^3 x_3], \quad \epsilon_S \sim \mathcal{N}(1, \sigma_S^2) \tag{5}$$

where $\epsilon_S \in \mathbb{R}^{N \times C}$ represents the sampled factors. $\sigma_S$ is the hyper-parameter that needed to be determined.

**MagW (Magnitude Warping)** Um et al. (2017). This method utilized a randomly generated smooth curve to corrupt the magnitude of the original signal. For each data channel, we sample $k$ nodes from $\mathcal{N}(1, \sigma_M^2)$, which results $knot \in \mathbb{R}^{N \times C \times k}$. Then, we interpolate $knot$ evenly with linear function and form $\epsilon_M$. The transformed view can be considered as:

$$\hat{\mathbf{X}} = \mathbf{X} + \epsilon_M \tag{6}$$

In this work, we fix $k$ to be 8 and only focus on the hyper-parameter $\sigma_M$.

**TimeW (Time Warping)** Um et al. (2017). TimeW distorts the time intervals between time steps to corrupt the temporal locations of the original signal. Similarly as the MagW, this method locates $k$ nodes by $\mathcal{N}(1, \sigma_T^2)$ and interpolate the data based on a randomly generated smooth curve. Same as our implementation of *MagW*, we linearly interpolate $k$ nodes to define $knot \in \mathbb{R}^{N \times C \times k}$ as the curve for TimeW. We fix $k$ to be 8 and focus on the hyper-parameter $\sigma_T$.

**Permutation** Um et al. (2017). Permutation is also a method for distorting the temporal locations of the original data. The permutation evenly splits the original signal into $K$ segments and shuffles the order of the segments:

$$\hat{\mathbf{X}} = shuffle([\text{segment}_1, \text{segment}_2, ..., \text{segment}_k]), \quad k \in [1, K] \tag{7}$$

where $K$ is the hyper-parameter to be determined.

## A.2   REPARAMETERIZATION

The aforementioned augmentation methods involve non-differentiable operations such as sampling values $S$ from certain distributions. For example, the Jitter method samples noise from a normal distribution of $\mathcal{N}(0, \sigma_J)$, which breaks the computational graph of the network. To propagate the gradients back to the $\sigma$, we applied reparameterization tricks to sample values for the augmentation operations. A randomly generated matrix $\epsilon$ with values from 0 to 1 is utilized.

**Normal Distribution**   The reparameterization of the Normal distribution can be considered as the following formula:

$$S = \mu + \sigma * \epsilon \tag{8}$$

Where $\mu$ and $\sigma$ are the mean and standard deviation values for the Normal distribution.

**Uniform Distribution**   The reparameterization of the Uniform distribution can be represented as:

$$S = l + (h - l) * \epsilon \tag{9}$$

Where $l$ and $h$ represents the low and high bound of the Uniform distribution.

**Bernoulli Distribution**   We used the RelaxedBernoulli distribution (Maddison et al., 2016) to approximate the Bernoulli Distribution:

$$S = \frac{\log(p) + \log(\epsilon) - \log(1 - \epsilon)}{t} \tag{10}$$

$p$ control the probability of drawing the positive samples, and $t \in [0, +\infty)$ represents the temperature, which controls the degree of approximation. When $t$ equals 0, the distribution is equivalent to the Bernoulli distribution; whereas if $t$ approaches $+\infty$, the resulting samples approach the constant value 0.5. In this work, we fixed $t$ to be 0.01.

## B  DATASETS

### B.1  PTB-XL:

The PTB-XLWagner et al. (2020) dataset is a large dataset containing 21,837 clinical 12-lead ECG records from 18,885 patients of 10 second length, where 52% are male and 48% are female with ages range from 0 to 95 years (median 62 and interquantile range of 22). There are two sampling rates: 100Hz and 500 Hz, available in the dataset. The raw ECG data are annotated by two cardiologists into five major categories, including normal ECG (NORM), myocardial infarction (MI), ST/T Change (STTC), Conduction Disturbance (CD) and Hypertrophy (HYP). The dataset contains a comprehensive collection of many different co-occurring pathologies and a large proportion of healthy control samples. To ensure fair comparison of machine learning algorithms trained on the dataset, we follow the recommended splits of training and test sets.

### B.2  APNEA-ECG

The Apnea-ECG Penzel et al. (2000) dataset studies the relationship between human sleep apnea symptoms and the heart activities (monitored by ECG). This database can be accessed through PhysionetGoldberger et al. (2000). This dataset contains 70 records with a sampling rate of 100 Hz, from where 35 records were divided into training, and the other 35 were divided into the test set. The duration of the records varies from slightly less than 7 hours to nearly 10 hours. The labels were the annotation of each minute of each recording indicating the presence or absence of sleep apnea. Thus, we split the ECG recording into each minute, which was a total of 6000 data points for each separation. We extracted 17233 samples for the training set and 17010 samples for the test set. And the ratio of non-apnea and apnea samples in the training set was 61.49% to 38.51%.

### B.3  SLEEP-EDFE

The Sleep-EDF (expanded) Kemp et al. (2018) dataset contains whole-night sleep recordings from 822 subjects with physiological signals and sleep stages that were annotated manually by well-trained technicians. In this dataset, the physiological signals, including Fpz-Cz/Pz-Oz ECG, EOG, and chin EMG, were sample in 100 Hz. To model the relationship between the sleep patterns and physiological data, we split the whole-night recordings into 30-second Fpz-Cz ECG segments as in Supratak & Guo (2020), which resulted in a total of 42308 ECG and sleep pattern pairs. We divided 25% of the samples into test set according to the order of the subject IDs.

### B.4  PAMAP2

The PAMAP2 Reiss & Stricker (2012) physical activity monitoring dataset consists data of 18 different physical activities, including household activities (sitting, walking, standing, vacuum cleaning, ironing, etc) and a variety of exercise activities (Nordic walking, playing soccer, rope jumping, etc), performed by 9 participants wearing three inertial measurement units (IMU) and a heart rate monitor. Accelerometer, gyroscope, magnetometer and temperature data are recorded from the 3 IMUs placed on three different locations (1 IMU over the wrist on the dominant arm, 1 IMU on the chest and 1 IMU on the dominant side's ankle) with sampling frequency 100Hz. Heart rate data are recorded from the heart rate monitor with sampling frequency 9Hz. The resulting dataset has 52 dimensions (3 x 17 (IMU) + 1 (heart rate) = 52). Following the same setting as used in Moya Rueda et al. (2018); Tamkin et al. (2020), We linearly interpolate the missing data (upsampling the sampling frequency from 9Hz to 100Hz for heart rate), then take random 10s windows from subject recordings with an overlap of 7s, using the same train/validation/test splits. As mentioned in Moya Rueda et al. (2018); Tamkin et al. (2020), 12 of the total 18 different physical activities are used in the experiments.

