# OpenReview forum: "Leaves: Learning Views for Time-Series Data in Contrastive Learning"
_ICLR.cc/2023/Conference — Submitted to ICLR 2023_

### Official Review · Reviewer_MoJJ · 2022-10-20

**Confidence:** 5
**Clarity, Quality, Novelty And Reproducibility:** The writing is clear. The algorithm i…
**Correctness:** 3
**Technical Novelty And Significance:** 2
**Empirical Novelty And Significance:** 2
**Recommendation:** 3

**Strength And Weaknesses:**

Strengths:
- The name LEAVES is really good
- Results for three of four considered datasets are better than baseline results
- The number of hyperparameters to learn for LEAVES is small
- The idea of using the reparametrization trick is nice (while natural)

Weakness, methods:
- According to the presented approach, we should select maximum distortion all the time. So, it is not clear, why this effect is not observed in practice, making the solution trivial
- The comparison and overall presentation of the method can benefit from looking on additional relevant papers that follow simiar ideas [3, 4]

Weakness, experiments:
- It seems, that to maximize the loss, we should distort the views with maximum values of sigma. It is true for most of the hyperparameters. Can you provide performance of LEAVES for hyperparameters that correspond to maximum distortion?
- What are the architectures for the considered datasets? Why they are selected in such way?
- Recent works on learning time series representations list particular important augmentations as well as ways to learn reporesentations to solve donwstream tasks [1, 2]. Why this work doesn't compare to those approaches and uses datasets different from that mentioned in these papers?
- It seems, that non-contrastive approaches are superior to SimCLR even for time series data [6]. The paper would benefit from adding this trainable view generator to e.g. BYOL similar to [3]
- Broader selection of the datasets for study would help to understand, what augmentations are useful in what settings. For example, it seems, that some of them can harm performance for periodic time series
- The ablation study doesn't demonstrate that all used augmentations are essential to the quality of the considered approach. In particular, does TimeDis contributes to the overall quality of the approach? Now it is not clear

Technical and presentation remarks:
- The exact training procedure is missing. Do we train the view generator and the encoder simultaneously? What other strategies are possible? Can other approaches improve the result?
- Figure 5: what is at the axes? Iterations or Epochs for x? Hyperparameter value for y?
- As the focus of the new method is performance, it makes sense to unite Table 1 - Table 4 in single Table in the main text, providing full version in appendix. It will help to save the space in the main text for desired technical details, that are missing now.

Literature:
1. Zhihan Yue, Yujing Wang, Juanyong Duan, Tianmeng Yang, Congrui Huang, Yunhai Tong, Bixiong Xu. TS2Vec: Towards Universal Representation of Time Series. AAAI. 2022
2. Eldele E., Ragab M., Chen Z., Wu M., Kwoh C. K., Li X., Guan C. Time-Series Representation Learning via Temporal and Contextual Contrasting. IJCAI. 2021.
3. Shi, Yuge, et al. "Adversarial masking for self-supervised learning." ICML. 2022.
4. Reed, Colorado J., et al. "Selfaugment: Automatic augmentation policies for self-supervised learning." CVPR. 2021.
5. Koyama, Masanori, et al. "Contrastive Representation Learning with Trainable Augmentation Channel." arXiv preprint arXiv:2111.07679 (2021).
6. Marusov, Alexander, Valerii Baianov, and Alexey Zaytsev. "Non-contrastive approaches to similarity learning: positive examples are all you need." arXiv preprint arXiv:2209.14750 (2022).

**Summary Of The Paper:**

The papers proposes to learn the view generation component for contrastive learning for time series data.
The alogithm is simple and includes usage of 5 common augmentations techniques and one new one (TimeDis) applied sequentialy to generate a view.
The loss function is similar to SimCLR and is maximized wrt parameters of LEAVES in an adversarial game.
The authors demonstrate improvement for the considered four medical datasets.

**Summary Of The Review:**

The approach seems interesting and simple, inheriting these qualities from a similar approach from CV ViewMaker.
While the obtained approach and results deserve attention, they provide a limited contribution to the community.

---

> ### Author Response · Authors · 2022-11-19
> **Response to Reviewer MoJJ**
>
> Thank you for your comments and suggestions. And many thanks for liking the name LEAVES, we like it, too.
>
> - It is true for most of the hyperparameters. Can you provide performance of LEAVES for hyperparameters that correspond to maximum distortion?
>
> According to the training curve, it is true that some parameters for augmentations converged to the maximum threshold we set. But if we use the max sigma value directly from the beginning, the encoder may not learn useful information. It is because we used an adversarial training strategy, thus, the degree of distortion of the learned views and the ability of the encoder keep increasing gradually during the training process.
>
> - Why this work doesn't compare to [1,2,3,4]?
>
> We totally agree with your point that there are many studies in the literature investigating contrastive learning on time-series data and achieving promising results. We will cite these studies in the revised manuscript. We did not include them as our baseline for the differences. For example, in [1] and [2] help the encoder with extracting latent representations in interesting ways;whereas our focus was on the augmentation part. Also, [3] and [4] are two very interesting studies that investigated auto augmentation in contrastive learning.  [3] aims to develop an auto-view method with only one type of augmentation, i.g., auto-masking images; [4] finds the optimal augmentation policy from the combination of various policies, which are different from our work. Again, we appreciate your comments on this point. We can board the horizon of our work by comparing and discussing the aforementioned methods and aspects.
>
> - It seems, that non-contrastive approaches are superior to SimCLR even for time series data [6]. The paper would benefit from adding this trainable view generator to e.g. BYOL similar to [3]
>
> Regarding the framework work, we used in this study. Currently, we are using SimCLR only, but we totally agree with the point that our method can be applied to examine to more other frameworks such as BYOL, MoCo, etc.
>
> - What are the architectures for the considered datasets? Why they are selected in such a way? Broader selection of the datasets for study would help to understand, what augmentations are useful in what settings.
>
> We selected several public datasets, and they were originally organized in different architectures. For example, the ECG-Apnea dataset was saved with consecutive hours of ECG signals with apnea labels every minute; PTB-XL already segmented data into 10-second segments with annotations. We selected the datasets because these databases are widely studied. And these data covered many types of the time-series data, such as irregular, periodic data, etc. We agree that using a broader selection of the dataset can help evaluate our method and understand the data augmentation in the time series domain. And we will consider more datasets in the future.
>
>
>
> - The ablation study doesn't demonstrate that all used augmentations are essential to the quality of the considered approach. In particular, does TimeDis contributes to the overall quality of the approach?
>
> Thank you for the suggestion. The ablation study for examining the contributions of different data augmentations is interesting. In our method, we aim to make the model learn the hyperparameters for different augmentation automatically. This means if specific types of augmentations are harmful or not essential to improve the quality of the learned representations, the parameters can be optimized to 0, which means the model would skip specific types of augmentation automatically. Thus, we showed the parameters learning curve as Fig. 5 to indicate the importance of augmentation instead of explicitly testing them individually.
>
> - The exact training procedure is missing. Do we train the view generator and the encoder simultaneously? What other strategies are possible? Can other approaches improve the result?
>
> Thank you for pointing out your confusion. We trained the framework using an adversarial training approach, which means the LEAVES (generator) and the encoder are trained simultaneously. Considering the various data augmentation methods we have, other strategies such as the mean teacher method may also be applicable in training our framework to avoid negative effects from improper augmentation policies.
>
> - Figure 5: what is at the axes?
> Yes, it is correct. X: interaction of epochs; y: values of hyperparameters
>
> - As the focus of the new method is performance, it makes sense to unite Table 1 - Table 4. It will help to save space in the main text for desired technical details, that are missing now.
>
> We listed 4 tables because we tried to compare our results with SOTAs on the same dataset, and many of them evaluated the performance in different metrics, which caused the difficulty of merging all of them together. But thank you for this suggestion. If possible, merge 4 tables together.

---

### Official Review · Reviewer_iTL4 · 2022-10-21

**Confidence:** 4
**Correctness:** 3
**Technical Novelty And Significance:** 2
**Empirical Novelty And Significance:** 3
**Recommendation:** 5

**Clarity, Quality, Novelty And Reproducibility:**

The novelty in this paper is a little bit low. Compared to ViewMaker, the novelty lies in the different types of augmentation to be learned. However, the types of augmentation used in the proposed method (Permutation, TimeDis, MagW, Scale and Jitter) are not designed delicately. It is possible that some are overlapped, e.g. Jitter and MagW, and other important types are missing, e.g. transformation in the frequency domain.

As mentioned above in “Cons” and “Questions”, the paper is not clear enough. The organization should be improved and the important details should be added. Without those details, the proposed method is not reproducible. Additionally, some words should be rephrased to avoid confusion. For example, in the title of 5.2, “fine-tune” has other meanings, i.e. updating the encoder in the downstream tasks, in that context.

**Strength And Weaknesses:**

Pros:
- Interesting topic. Contrastive learning has been shown to be very powerful, but it requires well-designed data augmentations, which has not been well studied for time-series data (compared to that for image). A good automatic time-series data augmentor should be beneficial for a great many real-world applications.
- Good results. This paper conducts experiments on four real-world time-series datasets, and the proposed method outperforms the supervised baseline and ViewMaker (an automatic domain-agnostic data augmentor) consistently. It is also very helpful to have SOTA methods listed in the tables.
- Nice ablation studies. The ablation studies show that the downstream performance for the four real-world tasks will be influenced by the choices of the hyper-parameters of data augmentations by a non-negligible margin, and the previous method, i.e. ViewMaker, is not able to generate satisfiable augmentations for time series data. Those underscore the need of designing a better automatic time-series data augmentor.

Cons:
- This paper only uses SimCLR as the contrastive learning framework to demonstrate everything. However, several contrastive learning frameworks (e.g. BYOL,  MoCo v2/v3) have been proposed after SimCLR and have been shown to outperform SimCLR empirically. How to adapt the automatic augmentation module into those frameworks (that should be easy for the proposed method) should be described, and whether the module can improve the performance of those frameworks empirically should be verified.
- The novelty of the method is a little bit low. The adversarial framework is the same as ViewMaker, with the only difference being the types of hyperparameters to be learned. Moreover, the proposed method can only learn the hyperparameters of augmentations that are already well-designed, and it can not generate new augmentations that have not been discovered yet. Therefore, using such a module still requires researchers to manually design augmentation types first.
- The organization of the paper can be improved. The definition and at least of a short explanation of the data augmentation used in the proposed module, i.e. Permutation, TimeDis, MagW, Scale and Jitter, should be included in the method section. Only from Figure 3, it is very unclear what they are. Moreover, the variables to be learned should be highlighted clearly, and why they are learnable (differentiable) should also be explained.

Question:
- What is the difference between MagW and Jittering? Isn’t MagW included in Jittering?
- In Equation (1), what is \sigma_P? Isn’t the variable to be learned for Permutation the number of segments K?
- Why are Permutation and TimeDis differentiable?
- How do you know when to stop during the training process? As shown in Figure 5, some of the parameters, like Perm Max Segments on Sleep-EDFE and on Apnea-ECG, are not converging eventually.
- Are the hyper-parameter values learned by LEAVES meaningful? Have you tried to re-train the encoder network using the (fixed) hyper-parameters learned by the adversarial training process? Will that deliver better or worse results compared to training the encoder and the hyper-parameters together?


**Summary Of The Paper:**

This paper tackles the problem of automatically generating augmentations for time-series data in the contrastive learning framework. It proposes LEAVES, a module that learns the hyper-parameters of the time-series data augmentations. The module is trained adversarially, i.e. maximizing the contrastive loss while the encoder network is minimizing the contrastive loss. Experiments on four public time-series datasets demonstrate the efficacy of the proposed method.

**Summary Of The Review:**

This paper addresses an interesting and important question - how to automatically generate good time-series data augmentations for contrastive learning. The empirical results of the proposed method are good, but the proposed method is not designed delicately and important details and experiments are missing. Therefore, the paper needs more work before it is ready to be published in my opinion, but I am happy to raise the score if all my concerns are addressed.

---

> ### Author Response · Authors · 2022-11-19
> **Response to Reviewer iTL4**
>
> We thank the reviewers for finding our paper interesting and our ablation studies nice.
>
> We appreciate the suggestions of trying other contrastive learning frameworks, such as BYOL and MoCo. The proposed method is mainly based on the SimCLR framework, and shows significant improvements. The SimCLR framework can be easily replaced by BYOL or MoCo framework, and we leave this as future work.
>
> Still requiring the pre-defined data augmentation methods is one of the limitations of the proposed method. Meanwhile, those pre-defined data augmentations can greatly help us generate more meaningful and realistic samples by serving as physiological constraints, in contrast to the random perturbation added to the visual image (as illustrated in Fig. 1). In addition to the constraints, the augmented data is generated through the composition of those pre-defined data augmentations, which can generate diverse while still meaningful examples.
>
>
> For the additional questions:
> - What is the difference between MagW and Jittering? Isn’t MagW included in Jittering?
>
> Both jittering and MagW are augmentations methods that distort the amplitude of data. The difference is: jittering adds noise to the signal randomly; whereas MagW first generates a smooth curve with the same length as the signal, then scales the signal according to the curve accordingly.
>
> -In Equation (1), what is \sigma_P? Isn’t the variable to be learned for Permutation the number of segments K?
>
> Yes, \sigma_P equals the number of segments K and indicates the parameters we desired to learn for the permutation augmentation.
>
> - Why are Permutation and TimeDis differentiable?
>
> Permutation: since permutation permutes the temporal location of the original signal, we make it differentiable by tracking the indexes of the permuted segments and permuting the gradients with the same indexes.
> TimeDis: the timeDis can be decomposed into 2 steps. (1) Generate a smooth curve with a gaussian mixture model. (2) Affine the original signal by the generated curve. (1) is differentiable by applying the reparameterization tricks. (2) We utilized the affine_grid function implemented by PyTorch to make it differentiable.
> We will show a short explanation about the data augmentation methods used in the method section in the revised manuscript.
>
> - How do you know when to stop during the training process? As shown in Figure 5, some of the parameters, like Perm Max Segments on Sleep-EDFE and on Apnea-ECG, are not converging eventually.
>
> This is a great question. We will also add this information to the next version of our manuscript. The target of the framework can be understood as finding the views with maximum distortion but are still distinguishable by the contrastive learning framework. Therefore, we stopped the training simply by checking whether the contrastive loss kept increasing after certain training epochs.
>
> - Are the hyperparameter values learned by LEAVES meaningful? Have you tried to re-train the encoder network using the (fixed) hyper-parameters learned by the adversarial training process? Will that deliver better or worse results compared to training the encoder and the hyper-parameters together?
>
> We appreciate your great question. So far, we haven’t done thorough experiments to investigate it. But empirically, also as mentioned in the previous question, the learned views already start to become undistinguishable by the contrastive encoders. Therefore, using the learned hyperparameters might generate excessive distortions on the signal, which causes the difficulties for encoders to learn at the beginning training stage.

---

### Official Review · Reviewer_KPs5 · 2022-10-25

**Confidence:** 4
**Correctness:** 3
**Technical Novelty And Significance:** 2
**Empirical Novelty And Significance:** 3
**Recommendation:** 5

**Clarity, Quality, Novelty And Reproducibility:**

Clarity: Clear.
Quality: Somewhat good. Well written, and extensive empirical evaluation on one hand. On the other hand, missing key baselines.
Originality: Derivative, to some extent. The proposed approaches have been shown to work well in other domains, such as computer vision.

**Strength And Weaknesses:**

Strenghts:
- The paper is easy to follow and the idea is clearly presented and explained.
- A diverse set of experiments are done to defend the claims of the paper. The model performs reasonably well against the proposed baselines.

Weaknesses:
- Differentiable augmentations are not a novel concept, and the adversarial framework to learn them is quite reminiscent of AdCo [1].
- One setup missing is the forecasting experiments in e.g. TS2Vec. This method could be directly applied to it, and results on a set of orthogonal tasks would be beneficial. More importantly, TS2Vec, CoST, etc. missing as (strong) baselines: comparing to them would strenghten the experiments section.


References:
[1] AdCo: Adversarial Contrast for Efficient Learning of Unsupervised Representations from Self-Trained Negative Adversaries


**Summary Of The Paper:**

The papers approaches the problem of contrastive learning of time series representations by learning parameters for time series augmentations. This allows good "views" of the time series to be directly incorporated into SimCLR, leading to reasonable performance gains against the baselines.

**Summary Of The Review:**

While I find this work well-written, and appreciate the extensive experiments, I also note the two main flaws I have listed above. Both those positive and negative points justify, in my opinion, the score I have given.

---

> ### Author Response · Authors · 2022-11-19
> **Response to Reviewer KPs5**
>
> Thank you for your comments and suggestions.
>
> Thanks for pointing out the Adco paper.We agree with the reviewer that the differentiable augmentations and adversarial training are not new techniques. However, the combination of them to time series data, especially biobehavioral data is not well studied. Directly adapting the method to time-series can generate highly-diverse but unrealistic samples, as shown in Fig. 1, which could potentially hurt the learning of meaningful representations. In this work, we resolve this problem and generate more reasonable views by well-designed augmentations as constraints from domain knowledge knowledge in time-series data.
>
> Although this study mainly focused on learning better data augmentation methods for time-series data in contrastive learning, we totally agree with the reviewer that some other methods, like TS2Vec and CoST,  are very interesting  and can be strong baselines. We will add the comparisons in the final version.

---

### Official Review · Reviewer_9y2Q · 2022-11-03

**Confidence:** 3
**Correctness:** 4
**Technical Novelty And Significance:** 2
**Empirical Novelty And Significance:** 2
**Recommendation:** 5

**Clarity, Quality, Novelty And Reproducibility:**

The paper has good clarity and good quality.

The novelty of this paper is acceptable but with low reproducibility since the datasets are closed.

**Strength And Weaknesses:**

Strength:
This paper proposed AutoAugment in their methods, which is quite useful for contrastive learning.

Weakness:
Their experiment just show a sole example on some medical case. It does not show a good contribution to the whole fields. Also their novelty is very empirical-based, I am not sure if this is suitable for this conference.

**Summary Of The Paper:**

In this paper, the authors propose LEAVES, which is a lightweight module for learning views on time-series
data in contrastive learning. The LEAVES is optimized adversarially against the contrastive loss
to generate challenging views for the encoder in learning representations.



**Summary Of The Review:**

In this paper, the author introduced a simple but effective LEAVES module to learn augmentations for timeseries data in contrastive learning. With an adversarial training manner, the proposed method optimizes the hyper-parameters for data augmentation methods in contrastive learning. The idea is OK but novelty is not very obvious.

---

> ### Author Response · Authors · 2022-11-19
> **Response to Reviewer 9y2Q**
>
> Thank you for your comments.
>
> As you pointed out, we mainly examined our method on human-centered or medical data in the manuscript; however, we will test our method with more general time series data in the revised manuscript.. We fully understand your concern about the scope of this work. But as our method is simple and very intuitive, we believe that it can be used for a wide range of sequential data for different applications.
>
> We proposed our method based on the limitation of the prior work on sequential data. As we described in the discussion section, the CNN-based methods, such as the ViewMaker, are computationally heavy and possible to introduce inappropriate noise into the sequential data. Our proposed method improved the prior works by introducing the domain knowledge of sequential augmentation with a very low cost in computation.
>
> Moreover, to add reproducibility, we evaluated our proposed method using 4 publicly available. We will also release our codes for processing these datasets later.

---

### Decision · Program_Chairs · 2023-01-20

**Decision:**

Reject

**Justification For Why Not Higher Score:**

- Lacks novelty
- Lacks clarity of presentation and organization

**Justification For Why Not Lower Score:**

The paper presents an approach to generate augmentations for Contrastive learning in time-series

**Metareview: Summary, Strengths And Weaknesses:**

The paper presents a simple approach to generate multiple views of time-series data augmentations towards contrastive learning. Though it has been demonstrated on SimCLR, it has potential for other contrastive learning methods too. The approach has been empirically evaluated on four public data sets.

The approach is simple and does not make a substantial contribution.